# Teen Perspectives on Suicides and Deaths in an Affluent Community: Perfectionism, Protection, and Exclusion

**DOI:** 10.3390/ijerph21040456

**Published:** 2024-04-09

**Authors:** Abigail Peterson, Carolyn Smith-Morris

**Affiliations:** 1Department of Anthropology, Southern Methodist University, Dallas, TX 75275, USA; 2School of Medicine, Case Western Reserve University, Cleveland, OH 44106, USA; 3Peter O’Donnell Jr. School of Public Health, UT Southwestern Medical Center, Dallas, TX 75390, USA; carolyn.smith-morris@utsouthwestern.edu

**Keywords:** adolescent, suicide, death, affluence, perfectionism

## Abstract

Clusters of youth suicide and death are tragic for communities and present long-term consequences for the surviving youths. Despite an awareness of community-based patterns in youth suicide, our understanding of the social and community factors behind these events remains poor. While links between poverty and suicide have been well documented, wealthy communities are rarely targeted in suicide research. In response to this gap, we conducted ethnographic research in a wealthy U.S. town that, over a recent 10-year period, witnessed at least four youth suicides and seven more youth accidental deaths. Our interviews (*n* = 30) explored community values and stressors, interpersonal relationships, and high school experiences on participant perceptions of community deaths. Youth participants characterize their affluent community as having (1) perfectionist standards; (2) permissive and sometimes absent parents; (3) socially competitive and superficial relationships; and (4) a “bubble” that is protective but also exclusionary. Our qualitative findings reveal network influence in teen suicides and accidental deaths in a wealthy community. Greater attention paid to the negative effects of subcultural values and stressors in affluent communities is warranted. Further, our work promotes the value of ethnographic, community-based methodologies for suicidology and treatment.

## 1. Introduction

Community trauma in the form of youth suicides and death produces ripples of harm that can reverberate for many years across a community. Indigenous and other communities experiencing historical trauma are particularly vulnerable to these patterns [1,2,3,4] but adverse childhood events (like the death of a classmate) may beget depression, coping challenges, and a loss of resilience [5] among youth in any given setting [6]. The social life of adolescents is, therefore, critical to understanding patterns in their mental health.

Despite an awareness of community-based patterns in youth suicide, our understanding of those social and community factors remains weak. Adolescents are known to be more vulnerable to suicide “contagion” [7] or “clusters” [8,9]. Yet, much of suicidology research is still focused on individualized treatment and post-trauma recovery, neglecting the community-level variables that may contribute to, stabilize, or fail to interrupt those patterns [10,11]. Accordingly, White and Kral have suggested that the study of youth suicide must attend to complex relational processes involving language, culture, and power [12]. Similarly, Abrutyn et al. call for a “rekeying [of] cultural scripts” as a strategy to interrupt youth suicide clusters [13]. In short, a robust community diagnosis or assessment is requisite to break these tragic cultural and relational cycles.

“Aurdon” (a pseudonym) is a wealthy U.S. town that, from the outside looking in, embodies the antithesis of a community “at risk”. College degrees and white-collar jobs are commonplace. At least one (but often two or more) luxury vehicle sits outside the family home. Vacations are routine, and expensive belongings are numerous. With that opportunity, however, comes a unique set of hazards. Over a recent 10-year period, the population of one of Aurdon’s two public high schools (herein referenced as “Crimson”) witnessed at least four youth suicides and seven more youth accidental deaths, six of which involved substance use. Links between poverty and suicide have been documented in multiple studies [14,15,16], but wealthy communities are so rarely targeted in research [17] that our understanding of adolescent suicide in affluent communities is limited. This is true despite evidence that U.S. youth in relatively affluent communities face both high achievement-driven pressures and relative disconnection from adults [18]. Certainly, high levels of criticism from adults, family, or peers can drive feelings of relative deprivation and emotional turmoil in teens of any SES level or cultural setting [4,19,20,21]. Still, relatively little work among affluent teens in their communities has been published. Other exceptions include Berger [22], who suggests that upper-class youth can face high achievement pressures, excessive parental criticism, and pressures toward perfectionism, and Mueller and Abrutyn [6], who suggest potential negative aspects of social integration for adolescents in achievement-oriented communities. Several authors have pointed to the relationship between perfectionism as a personality trait and suicide risk [23,24]. Perfectionism is seen to be shared among family members, and parents may predispose their children to this trait [25]. Furthermore, one’s social context and the expectations about one’s social network may also contribute to perfectionism and interpersonal stress [23]. Furthermore, in affluent communities with high social pressure and competition, mental healthcare may be stigmatized [26], especially among males [27], exacerbating a difficult period for youth development.

Edward Lowe, after studying the well-being of Micronesian youth, argued “it is instructive to examine the relationships among persons and socially organized domains of activity together” [28] rather than fixating exclusively on individuals’ psychological states or to social issues in isolation. This poses a challenge of course, as the intimate connections between individual experience, parental influence, and social factors in adolescent suicide are complex. How might social values and priorities shared within highly affluent communities create distinctive pressures on adolescents? How do teens who have abundant economic resources reflect on experiences of multiple, peer group deaths? To what factors do they attribute those clusters? Few studies provide insights into the high-stress, achievement-oriented communities as places to live and grow up or reveal what adolescents themselves narrate about these experiences (exceptions draw primarily from anthropological and Indigenous studies, such as Kral [29], Wexler [30], and Münster and Broz [31]).

To address this gap, we conducted an ethnographic study on young adult members of a highly affluent community designed to understand the effects of a spate of youth deaths and suicides. In particular, we conducted an ethnographic study among young adults (ages 18–23) who had lived and attended the public high school in Aurdon during the recent 10-year period in which at least four youth suicides and seven more youth accidental deaths (six of which involved substance use), occurred among students, young adult family members thereof, and recent graduates. Through an in-depth ethnographic study within this distinctive community, we aimed to explore subcultural values and stressors, the nature of interpersonal relationships, the role of high school experience, and the influence of community and parent relationships on participant views. It is with this objective in mind that the volume of youth deaths in Aurdon is comprehensively considered here rather than exclusively studying participants’ experiences with those deaths publicly known as suicide. The goal was to illuminate and learn from the lived experiences of young adults in Aurdon; participants were given the space to reflect comprehensively on their experiences without forcing discrimination upon them between known suicides and inadvertent deaths (whether correlated with high-risk behaviors or not). The interview process simultaneously allowed participants to weave their own narratives of meaning and researchers to gain insight from these narratives, drawing from the work of those like Owens and Lambert [32]. The sensitive and stigmatized nature of these topics being so great, adequate rapport was achieved through the ethnographic role played by Abigail Peterson whose relationship to this community is described below.

## 2. Methods

Our methodologies included long-term ethnographic engagement in this community over multiple years by Abigail Peterson; 3 months of observation over social media and local news during the months of and preceding data collection; open-ended conversation with 6 key informants on the topic of these recent deaths (namely, teachers, parents, and community members); and 30 semi-structured, recorded interviews with young adults (ages 18–23) who had lived and attended high school in Aurdon in the period during which 11 youth (accidental and suicidal) deaths had occurred. COVID-19-related protections (i.e., masking, social distancing, the use of Zoom for recorded interviews) were employed to protect both researchers and participants, as guided by the Southern Methodist University Institutional Review Board and the local county’s public health recommendations.

The observation of social media, the local news, and other public records were significant in providing researchers with a broad and informed perspective on community perspectives and reactions but are not cited here to protect identities. Similarly, many specifics about the young adults’ deaths are intentionally not provided in detail in this report for reasons two-fold: first, the purpose of this research was to understand young adults’ perspectives and reactions related thereto, and many details are irrelevant to the objective; second, with many of these deaths of a very-public nature, detailing specifics would inherently compromise community anonymity. The authors acknowledge the resultant limitations for readers but conclude that the findings are comprehensible and consequential notwithstanding this constraint.

The participants were recruited via social media (posted content was pre-approved by the Southern Methodist University Institutional Review Board) and word of mouth within the community. Participants were compensated for completed interviews with USD 15 Amazon e-gift cards. Notably, the recruitment process was incredibly easy for researchers and many participants noted that they “did not need compensation” to participate. Of course, they were compensated regardless of such statements, but the enthusiasm speaks to the significance of the topic in the lived experiences of participants.

Table 1 offers a summary of basic participant demographics. Note that additional options were available for participants to self-identify gender, race/ethnicity, etc. Table 1 shows only those options that were selected.

Interview prompts were open-ended and designed to allow participants (named throughout this report with pseudonyms) to identify the priorities and characteristics of Aurdon that they felt were most relevant to the topic. See Table 2 for a sample of prompts used. Care was taken to avoid introducing concepts from the suicidology literature.

Recorded interviews were transcribed verbatim, then coded and analyzed using Dedoose 7.0.23 software for ethnographic data analysis [33] to identify recurrent themes, categories, and expressions. Coding occurred in two steps: interviews were initially coded for recurrent, explanatory, or descriptive themes, with Inter-Rater Reliability scores taken after coding by each author of 2 interviews, and again after 4 interviews, achieving an IRR of 0.81. See Table 3.

Upon the conclusion of the coding process, authors worked to reconcile findings and organize resultant threads for dissemination.

## 3. Results

Aurdon is a community with a per capita income of just over USD 100,000, a median value of homes over USD 2 million, and 68% of the majority White population holding a bachelor’s degree or higher (U.S. Census). Luxury possessions are commonplace. Those attending the public high school in question have access to resources often unavailable to public school students, with advanced placement courses, competitive sports teams, and the financial ability (for many) to hire private tutors among them. On standardized state exams, students outperformed both district and state averages in all three sections: mathematics, reading, and science. Attending college is expected after high school graduation. While these generalizations do not accurately describe all families in the community, the general culture is captured in this description as a foundation for contextualizing dialogue with participants. For ease and clarity, the geographic community of Aurdon and the community of people associated with the public high school Crimson are treated as synonymous entities and are referred to as such throughout this report.

Allow us to introduce a few participants to represent the group. First, we present Gracyn. Gracyn spent the majority of her young life growing up in Aurdon and attended all four years of high school there. At the time of her interview, she was a senior in high school. Gracyn’s parents were (and are) loving, involved in her life, and wealthy enough to fit into Aurdon’s financial standards seamlessly.

Next, meet Kian. Kian’s family income was well below average in Aurdon, and their modest means meant that he grew up with far fewer luxuries than were normalized amongst his peers. Kian had graduated from high school a few years prior to his interview and was working full-time. Kian did not attend college.

Finally, we introduce Ace. Ace was a freshman in college at the time of his interview. A young man with a warm demeanor and a positive attitude, he reported a strong sense of satisfaction with his high school experience when reflecting on his activities and academics.

Three major themes emerged from the thirty interviews conducted, which we discuss in turn below. Some participants are quoted multiple times and others are not at all; excerpts were selected to best illustrate group-wide themes.

The first major theme developed through participant narratives about the community of Aurdon, including their general perceptions of this area as a place to grow up, and particularly in which to attend high school.

### 3.1. Youth Views on the Affluence of Aurdon

Informants described Aurdon as not only wealthy but also as visibly and extravagantly wealth-centered. This was captured through the code “affluence”, applied 194 times across 28 transcripts.

Simon: If you grow up in that, …you can kind of grow up with a certain sense of entitlement or like a certain sense of naivete, because all you’ve all you’ve known is the best thing in the world. So I’m kind of glad that I wasn’t necessarily born there.

Jacob: I could say the stereotype for our community is pretty well-known, in being the physical-looking, fake-boobs wife who kind of, has a super-rich husband, drives a Range Rover. I mean, that’s… pretty true.

Chloe: I feel like a lot of people were very materialistic and, like, really didn’t care about, I don’t know, the most important things in life.

Roland: You have to be excessive and extravagant to set yourself apart.

Participants widely acknowledged this atypical level of wealth in Aurdon. And, while participants recognized there was a mix of income levels in the community (e.g., Aria and Jacob both acknowledged this diversity), the topic of wealth and its everyday influence on their experience pervaded discussions about growing up here. The contrasting narratives below, by Fabian and Kian, reveal this diversity:

Fabian: And so, you know, obviously a lot of these kids are very spoiled, myself included…. All these kids get brand-new cars for their birthdays, and, you know, they fly to Cabo for Spring Break. You know, it’s like they live a life of luxury and that culture really kind of permeates at school.

Kian: So it would just suck when like, I see… every single one of my friends go home and … are able to study for all hours in like this, their own room… I lived in—a one bedroom apartment. I was in the, like, living room of that apartment.

Aurdon’s wealth revealed itself in less explicit ways, as well. Greer, for example, was not distressed because she did not have access to a car, but because she was embarrassed by the *quality* of her car:

Greer: What I did not love about high school, was—probably, like, the sense of pressure. And this is, I think, just for girls, because people are so materialistic in [Aurdon], I think, sometimes, like that was a constant pressure like “what am I going to wear today?”. My first car, this sounds silly, but my first car was like a really beat-up, old, [mid-range SUV], and I just remember being, like, so embarrassed to drive that around campus and, like, kind of wanting to park in the back, because my best friends had, like, a new car whatever.

In brief, ethnographic interviews clearly established a shared experience of Aurdon’s wealth and characterized the ways that the community’s general affluence impacted different youth. We elaborate on those details next.

### 3.2. A Community of Perfectionists

A second descriptor for Aurdon, often co-occurring with “affluence”, demarcated a cluster of traits or efforts associated with achieving this level of affluence, and which earned the code name of “perfectionism”. As this term is employed in the suicidology literature [34,35,36]), “perfectionism” is a multidimensional trait that involves high personal standards for achievement, concern over mistakes, and high parental expectations and criticism, all of which may produce maladjustment, helplessness, or hopelessness. (In particular, the work of Hewitt and Flett [36] identifies differences between self-oriented, other-oriented, and socially prescribed perfectionism, thereby confirming that different sources of pressure exist and may overlap within an individual’s sense of their standards.) Our coding captured narratives on these characteristics as well as participant remarks about pressures toward perfectionism that might come from parents, peers, or the broader community (see, e.g., Frost et al. [34]).

The code “perfectionism” formed a major code set in our research (with 371 code applications across 30 interviews) marking passages that addressed achievement pressures and high expectations on young adults; commentaries about drive and competitiveness; and the set of high standards to be upheld on everything from academic and athletic achievement to social popularity and wealth.

Harrison: I think it’s just an inherently competitive environment, right, because you have really well-off people that are basically starting life on third base.

Joseph: There’s pressure in any, every community, and… I think it’s a good thing. …I think pressure creates success.

Summer: [If] you don’t major in business or something, you know, expensive, that’s built to make money, then you’re going to be useless in our area… Like it has to be, like, incredible or you’re useless.

Participants acknowledged the toll these achievement pressures could take on a person’s mental health:

Bree: There’s lots of keep up with. Like, whether that’s like, materials, or like, cars and, like, wallets or shoes, you know what I mean. I think that’s a lot of pressure, which might not be the most challenging thing, but I think mentally, that takes a big toll. …It’s a lot to keep up with. Um, so that’s so, materialistically. And I think socially, partying started in seventh grade and if you didn’t really do that, …like people were cool about it, but like at the same time, you’re not going to get invited to stuff on the weekends….

Ace: It’s like a, it’s like a bubble…. It’s like people try to create this perfect norm that you’re supposed to be perfect and that everything’s great and happy. But it’s not. It’s life. Things happen.

Aria: [T]he adult community isn’t very welcoming, you know? And it has this very competitive environment. Like there’s always the need to compete with other people, and if you’re not at a certain level you’re judged, you’re looked at differently.

Noteworthy among the Aurdon narratives on social and academic pressures were remarks on the constancy of this pressure:

Riley: I think that there was a lot of pressure on kids all the time to perform, to outperform, the person to the right and to their left, all the time.

Jones: I feel like a lot of kids have a lot to look up to, or a lot to achieve, you know? They feel like they can never achieve what their parents have done.

Dana: I do remember always feeling that, like, pressure with the competitiveness of social life, of academics, of sports, like—kind of feeling like you have to check all of the boxes.

Gracyn: There’s social pressures to dress a certain way, act a certain way, live your life a certain way, have your sexuality be a certain way… It’s very conformist, I think… And yeah, like there’s just social pressures to do this when you go out, to be friends with this person, to post this on some random app.

Not only were the standards in Aurdon exceptionally high, but some also felt as though the standard kept moving; satisfaction with one standard did not result in a celebration but another set of goals:

Joy: There was always a pressure to do better and to be better.

Kian: You feel like you’re never enough.

While perfectionism was occasionally described as a positive trait or a motivating force, participant narratives most often conveyed a sense of discouragement over failures or mistakes. Corresponding to established definitions of the perfectionist trait, participants viewed setbacks as socially unacceptable or even shameful.

Contributing to participants’ personal sense of disappointment or shame over mistakes, parents were said to hide their children’s mistakes and failures to maintain an image of perfection.

Joy: The parents here try and cover up their kids’ mistakes… so it didn’t affect the kids’ future… It’s like half and half. Like, most [parents] are “learn from your mistakes” but there are definitely some that, again, the competitiveness drive, a lot of this just comes back from competition. [They] will try and cover it up and just be like, “my kid didn’t do it” or, “denial”, “denial”, “denial”, and then if it does come out, do everything that you can to bury it.

Riley: I think that a lot of times they feel like they can fix their kids’ mistakes for them, without the kids figuring it out on their own, or letting the kids kind of deal with it and grow and mature from it.

Lacey: Kids [here] tend to get off easy sometimes.

Thus, parents—in attempts to encourage and protect their children—encouraged perfectionism and added to adolescent pressures. For example, wealth provided parents with the ability to “cover up” a child’s errors and to remove accountability:

Joy: A lot of places, their parents would just be like, “Okay. You screwed up. Like, you gotta deal with the consequences. We’ll help you, like, come back from that, but, like, sorry”. But here, I think,... [parents] will try and cover it up and just be like, “my kid didn’t do it” or, “denial”, “denial”, “denial”. And then if it does come out, do everything that you can to bury it.

Lacey: I do think, just because there is a lot of privilege, that ... like a bigger mistake, like doing drugs at school… that gets covered up. Or like, the cheating scandal [when a handful of students were hacking school computers to change their grades] … oh all those kids like got their records sealed and got into really good schools. And it’s like that’s because of the privilege.

Far from seeing the harm in these parental acts, adolescents who were protected from their mistakes, or who witnessed others being protected in these ways, felt safe, even “indestructible”, as Peter explained:

Peter: You could argue that there was this… feeling of indestructibility amongst our younger youths, because it’s almost like “anything I do will be fixed by my parents”, almost.

Bree: It’s almost like the limitations of helping your kids succeed—it is limitless.

But, in this context of both great material wealth and unlimited support from parents, participants perceived there to be “no excuse for failure to excel” or “to be sad”. Gracyn describes:

Gracyn: There’s just an overall stigma about mental health here, where it’s like, when you’re in such an affluent community it’s—“How can you be upset? How can you be sad?”

We asked how normal struggles, including mental health struggles, were addressed by youth in this context of wealth, competition, and perfectionism. In response, participants described mental health problems as potentially shameful and socially taboo.

Fabian: It’s super superficial. It’s all about how you’re posturing… Everybody is so concerned with their appearance and what they look like on the outside that there’s no talk about what’s going on on the inside, which is mental health.

Reggie: I definitely think vulnerability was lacking.

Robyn: [People here are] fake, putting on a front for other people, wanting to always look their best, just not being, like, raw, or real, or having real emotions.

Notably, multiple participants acknowledged that their interview was the first time they had a space to speak in depth about their thoughts and feelings in response to community (and in many cases, personal) tragedy.

In sum, narratives about perfectionism reveal how young adults felt high and constant pressure to achieve academic and monetary goals and saw failures or mental health struggles stigmatized within the community. These perceptions left young adults with conflicting, coexisting feelings of gratitude—for the privilege and rigor that prepared them for the world after high school—alongside feelings of frustration and stigma—for their upbringing in a community with such narrow definitions of success.

### 3.3. Protective but Exclusive

As speakers reflected on Aurdon’s spate of youth deaths and suicides—young people with whom they had gone to high school—participants attempted to make sense of reckless, dangerous, or suicidal behaviors. Participants named both community-level factors in youth deaths—such as community wealth and competitiveness—as well as parental influences. In particular, several youths told stories of parents who allowed unsupervised parties and tolerated adolescent substance use, which was reportedly both common and excessive:

Fabian: Like, I went to lots of parties in lots of places where the parents weren’t there, and there were just kids trashing the place and drinking a bunch of alcohol. At some point it was like, what is really going on here? It’s like this beautiful house with a hundred 17-year-olds in it. It’s kind of weird.

Chloe: There were so many parties that I had gone to where the parents were gone for the weekend and, ya know, people would get into their liquor cabinet and bring cases of beer and handles of liquor. And I just, like, I always thought that was normal.

Our participants suggested the high number of deaths emerged from a combination of extreme affluence, a hyper-competitive atmosphere, and the access adolescents had to alcohol and other substances. In particular, this combination was blamed for creating a community that was disconnected, selfish, and rule-breaking:

Pam: It’s a bubble. And people are aware of it… It’s like a gilded yacht cage… You keep it by forming your group and upholding your standards and keeping yourself together, and you don’t let a whole lot of other people in.

Dana: I was just, honestly, constantly disappointed in, like, the lack of humanity that I saw.

Fabian: But it was just sort of like closed-mindedness, or I don’t know if there was an ignorance or a lack of caring about the outside world, but [people were] very removed.

Roland: I think while there was some of that at [Crimson]… portraying oneself as like… “I’m not only wealthy but I’m also going to be a rulebreaker”.

While most of our informants seemed to exist somewhere inside the bubble they described, a handful of participants expressed feeling excluded:

Robyn: I feel like a[n Aurdon] mentality is very much being selfish and just really thinking about what you can do for yourself, and not really others.

Aria: I’ve also met parents who just—they’re just not very nice. …parents are very much like, if they see you, they might see you as a threat to their kid’s success.

Summer: Socially, I had, like, a few good friends, but I never felt like this was the best place to create relationships, just because of everything that was in the way. Whether that be what you wear, what you drive, where you’re going on summer vacation. I always felt like those things were kind of always in your face.

Riley: I would say that the culture was not great within the community. And it was really hard for kids to find, like, a really, really solid friendship group within high school; to have that social support during those hard times.

So, participants blamed the community-wide focus on wealth and social competition as barriers to developing close personal relationships:

Fabian: It’s all about, you know, celebrity and do you have the newest car, and like who have you hooked up with, and who is the hottest girl in the school. You know, it’s a very superficial culture.

Of course, we acknowledge—as did a few participants—that Aurdon’s exclusivity was not universal, and there are “a lot of great people that are willing to help”:

Joseph: I think, um, [Aurdon] as a community of being, um, wealthy and pretentious, but if you’re able to—see through that, there’s a lot of great people that are willing to help.

Reflecting on Aurdon’s exclusivity and competitiveness, young people were aware of the fact that this might one day apply to them. If they failed to earn the high income required to live in Aurdon, they might be unable to live (as independent adults) in this community. Participants described how this realization affected both young people and parents:

Bree: I think it’s seeing your parents’ success…. Once you grow up experiencing that type of lifestyle, subconsciously, you want to provide that for your [own] kids, or you feel pressure to continue that legacy. Um, so—I know this is harsh, but I’d say it’s the money that puts a subconscious pressure [on Aurdon youth]…

Pam: The perception of the community is why would you ever want to leave? …Nobody thinks, “...it’s okay that I won’t make enough money to live here”…. Because all the things feel like a pyramid, right, or like a race. And if it’s a multigenerational thing, what you are really running is a relay race.

Joy: Being really competitive, I think that comes from [parents] wanting to have, their kids to have, a really successful life and be back in [Aurdon], but—forgetting the fact that they didn’t start in [Aurdon].

In other words, Aurdon’s exclusive bubble could exclude the very children its cultural values had worked to produce: wealthy and high-achieving children of current residents. It is noteworthy that most participants specifically referenced one young man’s suicide over the course of their interview, for that young man referenced the community’s social and achievement pressures in a suicide note. These youth faced a looming future that would exclude them from their own hometown for the very reasons—competitiveness, affluence, perfectionism—that had been central to their upbringing and identity.

Gracyn grew up entrenched in these realities of Aurdon. At the time of her interview, Gracyn had already been granted admission to an Ivy League university; a model of achievement among Aurdon’s young adults, Gracyn worried for the future of her community’s young adults. One moment in her interview summarized this feeling:

Gracyn: …the community is not helping students in the right way. It’s not fostering, like, an environment of growth and openness. It’s very much about “this is your own personal struggle; deal with it. Don’t bring it to me”. And so, I mean, obviously, it’s not a helpful thing. It’s not working well for students. We had a very big example of such [(the very-public suicide and related note of the aforementioned young man)], and yet it still persists.

When Gracyn was a freshman in high school, one of her peers died by suicide. At the time of her interview, she reported that she thinks about it at least four times every week.

Kian moved to Aurdon just in time for high school; he did not grow up in Aurdon, did not attend middle school in the community, and did not have a family life that matched the Aurdon “ideal”, financially or otherwise. His high school friends, so he told us, lived similar lives amongst what he coined the “lower class” of Aurdon, and his options for making friends, so he felt, were limited to that “class”: those that could not afford the expensive outings and hangouts that others could, those that had to work in high school, etc. Throughout his time in Aurdon, Kian felt judged:

Kian: It felt like you were always being watched… someone was always telling you that you can do better, no matter how good you did.

About a year prior to his interview, Kian learned that one of his friends from high school had overdosed.

Ace had lived in Aurdon his entire life and grew up in a supportive and stable household. Ace’s account of his community coming together in the wake of loss to support each other was truly beautiful, yet we could not help but be disturbed by the phrase he used to describe the volume of loss experienced:

Ace: …for me, and for a lot of, for a lot of my friends, it kind of felt like groundhog’s day. We were just, kind of, repeating the same thing that happened…

While Ace was in high school, he was exposed to the death of a classmate after a fatal car accident, the suicide of another classmate, and the suicide of his best friend’s brother. Just months prior to his interview, one of Ace’s friends from high school died after an incident involving drug use.

Ace: We kind of already know what happens, how to react, and how to be there for one another, so it’s not really, like, a shock or surprise anymore, which is really, really sad but that’s honestly just the, kind of, the hard truth.

## 4. Discussion

The community of Aurdon has experienced considerable youth suicide and drug- and alcohol-related deaths in recent years. We questioned this sample of youth without employing terms from the suicidology literature, simply asking participants to describe their time in this community, what they knew of recent teen deaths, and how it had affected them. It was noteworthy that young adults living in this affluent community characterize Aurdon as having four characteristics well recognized within the suicidology literature as correlating with adolescent suicide: (1) perfectionist standards; (2) parents who are permissive and sometimes uninvolved or absent; (3) socially competitive and, for some, superficial relationships; and (4) a “bubble” that is protective but also exclusionary.

First, while a few participants acknowledged Aurdon’s perfectionism to be a motivating factor, the perfectionism and competition of this community were predominantly viewed as creating a discouraging atmosphere, as stigmatizing of mental health struggles, and as standing in the way of close relationships. The presence of these community characteristics within a setting of high adolescent suicide and accidental deaths (frequently involving substance use) corroborates the substantial suicidology literature in linking teen suicide to perfectionism [23,24,25]. Thus, our narrative findings confirm but elaborate on the character, content, and perception of perfectionism in this community. Indeed, we would argue that “perfectionism” must have a particular target or goal, shared by a network or community, for social pressures to be effective. For example, our participants frequently linked perfectionism to wealth or displays of wealth, such that achieving this distinctive level of wealth and locally valorized displays were crucial to notions of perfectionism. In short, traits of “perfectionism” are less informative of a particular or singular outcome and more reflective of rigid requirements in productivity or performativity (e.g., White and Morris [37,38]).

Second, our findings align with those of Singh and Behmani [39] who found permissive or uninvolved parenting to be associated with suicide ideation in youth. The perceived permissiveness of Aurdon’s parents for their adolescent substance use and unsupervised (or blind-eye-turning) parties was a broadly recognized characteristic. Permissiveness was not only described as a reward for adolescents’ hard work but was also described as the result of parents’ relative disconnection, physical absence, or emotional distance. Of course, it should be noted that parenting varies widely and with significant nuance; permissiveness did not always reflect a lack of involvement or emotional engagement, pressure does not always come without simultaneous support, and a generalization about community parenting oversimplifies variation between homes. That said, the significance of parent involvement and/or lack thereof was notable from participant interviews and, therefore, worth addressing. The youth in our study did not blame parents’ permissiveness for these deaths but did suggest a connection between parenting styles and adolescent risk-taking or rule-breaking behaviors. Thus, while our findings support research linking parenting style to both perfectionism [40] and suicide [41], we argue that a better understanding of this relationship is captured when a wider context—including familism [42,43], neighborhood [44,45,46], and social support [47,48]—is taken into account.

Third, our participants varied in the degree to which they could identify close and long-lasting relationships from their high school years. For those who knew peers who had died, doubt in the genuineness of their friends may reinforce feelings of loneliness, distrust, and isolation. On the other hand, many of our participants acknowledged close friendships through sports teams. However, all commented on the superficiality of some relationships among peers trying to maintain a reputation for friendliness. The protective role of positive teen relationships has recently been reviewed by Roach [49]. Further research, ideally involving a network analysis of bonds between adolescents [50], could shed light on the impact of deep vs. superficial relationships on teen suicidality.

Our fourth area of findings reflects the fairly novel focus of our research on an affluent community. The wealth of this community is visibly evident in markers of status and trends, and in social behavior and values that celebrate wealth as evidence of one’s achievements. But the wealth of Aurdon that opened certain doors for these adolescents is accompanied by high expectations and significant pressure. Although these teens had access to support services, tutors, and other costly resources, most viewed wealth not as a protective factor but as something that characterized local forms of competition and perfectionism. Also, experiences of living in a protected “bubble” or of being “invincible” thanks to one’s parents’ ability to hide or “erase” mistakes may lead Aurdon youth into risky behavior. Wealth, in this circumstance, was viewed as toxic rather than protective. Taken together, participants’ uncertainty about the authenticity and depth of their relationships and the counter-intuitive “toxic” results of access to resources evoke important considerations for *perceived* social capital as it relates to well-being. The existing literature has demonstrated that well-being must be considered a multidimensional construct [51] and our findings add another layer to that definition by suggesting that the construct must be considered with contextual nuance.

These thematic, ethnographic findings not only lend detail and character to our understanding of common correlates of teen suicide but also complicate our understanding of access, protective resources, and collective influences on the mental health of teens. This relatively unique study of teens in an affluent community points to a need for more collective intervention strategies that engage peer groups, parents or family units, and communities as a whole. Important strides have been made in recognizing the multiple levels of factors (e.g., individual, family, community) in teen suicide [8,28,38] and mental health [52]. Community coalitions are considered best practice for suicide containment and to disrupt the emergence of suicide clusters (see, e.g., Hacker et al. [7]). Such coalitions are broadly designed to identify at-risk youth, grow and deploy local support and intervention resources, catalyze teen leadership efforts, and develop targeted educational materials for youth and parents. Yet, the substantial cultural shifts that would decenter perfectionist and competitive values across a community may be quite deeply rooted, and, therefore, slow to respond.

Limitations of this study include the small sample size of participants and barriers created by the protection of community confidentiality. The use of ethnographic methods was an intentional decision made to elicit nuanced, individualized data from participants about a sensitive and subtle subject matter. That said, generalizability is always influenced by smaller sample sizes. This fact reinforces the need for further research. The sensitivity of the topic also necessitated strict protections for both participant and community anonymity. As acknowledged previously, the authors concede that additional detail about the community and circumstances of youth death and suicide would provide greater context for readers and future researchers, but they believe the findings to be impactful notwithstanding this limitation.

## 5. Conclusions

Effectively addressing adolescent mental health requires ongoing attention, since traumas like peer deaths can be continually renegotiated and their meanings can change over time [53]. When the influence of social structures and values on teen suicide clusters is considered [54], interventions to address youth suicide risk may require deep community and parental participation.

Our ethnographic results point to a greater community or network influence in teen suicides and accidental deaths than dominant treatment models suggest. For teens in this affluent community, the interplay of parental style, community-wide competition and perfectionism, and adolescent risky behaviors and feelings of isolation were central to the pattern of recent deaths. Greater attention to subcultural values and stressors, as well as the nature and influence of community and parent relationships in participant views, may be the key to effective suicide prevention. And, affluence cannot necessarily be viewed as a protective factor.

## Figures and Tables

**Table 1 ijerph-21-00456-t001:** Participant Demographics.

Participant Demographics (Self-Identified)		Count
Gender	Female	16
	Male	14
Age at time of interview	Age 18	4
	Age 19	4
	Age 20	2
	Age 21	14
	Age 22	5
	Age 23	1
Race/Ethnicity	Asian	1
	White/Caucasian	25
	Mixed Heritage	4
Years spent attending high school in Aurdon	4+	30

**Table 2 ijerph-21-00456-t002:** Open Interview Prompts.

Open Interview Prompts
1. Tell me a little about your high school experience here in [Aurdon]. 2. Were you aware of young adult or student deaths during your high school years? If so, can you tell me how you experienced that? 3. Why do you think your community has lost so many young people in recent years? 4. What was the most helpful or impactful thing said or done to help you cope with loss/grief? 5. Has the community changed after these losses? (If so, how?)

**Table 3 ijerph-21-00456-t003:** Coded Themes.

Code	Total Count
**Affluence** *Pressure, support, bubble*	194
**Apology** *Apologies, guilt*	55
**Community** *Community, culture, neighborhood*	628
→ **Bubble** (Axial code) *Participant use of the word or obvious allusion to the idea*	(14)
**Family** *Parent–child parallels, parenting, household dynamics*	495
→ **Permissiveness** (Axial code) *Parents covering up mistakes, parents liberal with substances*	(69)
**Fatigue** *Numbness, normalization, desensitization vs. familiarity, hopelessness, weariness, capacity for change*	82
**Grief** *Memory, trauma, grief, gender differences therein, fragility; includes reaction to death; includes modality of participant learning about community tragedies*	354
**Image** *Authenticity vs. superficiality, image, perceptions, performance/performative behaviors, stigma*	274
**Key Support** *Role model, best friend, friend group (tight)*	275
**Noteworthy** *Particularly poignant or summative*	296
→ **Gender Differences** (Axial code) *References/statements*	(55)
**Perfectionism** *Define/story, maxing out your effort, doing extremely well, college on a pedestal, equating college with personal value/worth; competitiveness/competition*	371
**Posting** *Social media, online “performance”*	89
**Prevent** *Preventability*	221
**Temporality** *Temporary vs. lasting, cyclical, changes in perspective with time*	198

## Data Availability

Due to the sensitive nature of research and ethical prioritization of participant anonymity, data is not available publicly.

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
