# Peer review of "Teen Perspectives on Suicides and Deaths in an Affluent Community: Perfectionism, Protection, and Exclusion"

_ijerph, 2024, doi:10.3390/ijerph21040456_

Round 1

Reviewer 1 Report

Comments and Suggestions for Authors

Appreciated,

I am sending you the review report.

Yours sincerely

Author Response

Thank you for reviewing this article and providing your feedback. The following edits have been made in response to your comments:

  • The abstract has been edited for further clarity.
  • A paragraph explicitly outlining limitations of the study has been added to the end of the Discussion section.
  • Further context about the community (lifestyle, etc.) has been added to the introduction section.
  • Additional information about participant compensation and motivation for involvement has been added.

Reviewer 2 Report

Comments and Suggestions for Authors

It is an interesting ethnographic study of suicide and accidental deaths in an economically prosperous community.

The abstract is appropriate for the content of the manuscript.

The introduction correctly points out the lack of knowledge about suicidal behavior in affluent communities. Consider it important to include the community suicide rate or the approximate number of youth in schools to weigh the number of suicides and accidental deaths. It is also important to consider the availability of mental health services in the community or nearby communities and their accessibility. It is reported that Inuts in Canada have higher suicide rates, due to difficulty accessing mental health services, among other factors.

The methodology is clearly described.

The results are clear and show the participants' statements clearly and transparently.

The discussion is relevant to the findings. Consider that an aspect omitted from the discussion may be the social stigma of the community associated with presenting mental problems, caused by their perfectionism. The social stigma to ask for help for emotional distress can be an important aspect that favors the appearance of suicidal behavior and addictions. It is important to include the possible limitations of the study.

The conclusions are supported by the findings.

Author Response

Thank you for reviewing this article and providing your feedback. The following has been addressed in our piece in response to your comments:

  • A new paragraph has been added to the end of the discussion section to directly address study limitations. Some limitations were noted throughout the piece but we agree that it helps to have a discrete section addressing limitations in the discussion.
  • Social stigma is certainly an important barrier to seeking support for mental health. We believe this is appropriately acknowledged in lines 337-338, 350-352, and 500-503.

Reviewer 3 Report

Comments and Suggestions for Authors

Well-delineated and conducted research. Socially remarkable objectives. Results presented clearly. Discussion sufficiently grounded. I suggest two modifications in the Abstract, so it points out the core of the main research question and grows interest in future readers/researchers: 

  1. To replace the last paragraph (lines 21-24) with what is at the end of the Conclusions: “ Our ethnographic results point to a greater community or network influence in teen suicides and accidental deaths than dominant treatment models suggest. For teens in this affluent community, the interplay of parental style, community-wide competition and perfectionism, and adolescent risky behaviors and feelings of isolation was central to the pattern of recent deaths. Greater attention to subcultural values and stressors, affluence cannot necessarily be viewed as a protective factor.”
  2. Discard the keyword “qualitative methods” because it adds no substantial information neither for the study nor for the search mechanism of data bases.

Author Response

Thank you for reviewing this article and providing your feedback. The following has been addressed in our piece in response to your comments:

  • We modified the end of the abstract to include important points from the conclusion excerpt you suggested. Due to word count restrictions, it is a modified version rather than a direct insert.
  • "Qualitative methods" has been removed from the keywords